# Large-Scale Contextual Market Equilibrium Computation through Deep Learning

## Abstract

Market equilibrium is one of the most fundamental solution concepts in economics and social optimization analysis. Existing works on market equilibrium computation primarily focus on settings with a relatively small number of buyers. Motivated by this, our paper investigates the computation of market equilibrium in scenarios with a large-scale buyer population, where buyers and goods are represented by their contexts. Building on this realistic and generalized contextual market model, we introduce MarketFCNet, a deep learning-based method for approximating market equilibrium. We start by parameterizing the allocation of each good to each buyer using a neural network, which depends solely on the context of the buyer and the good. Next, we propose an efficient method to estimate the loss function of the training algorithm unbiasedly, enabling us to optimize the network parameters through gradient descent. To evaluate the approximated solution, we introduce a metric called Nash Gap, which quantifies the deviation of the given allocation and price pair from the market equilibrium. Experimental results indicate that MarketFCNet delivers competitive performance and significantly lower running times compared to existing methods as the market scale expands, demonstrating the potential of deep learning-based methods to accelerate the approximation of large-scale contextual market equilibrium.

## 1 Introduction

Market equilibrium is a solution concept in microeconomics theory, which studies how *individuals* amongst groups will exchange their *goods* to get each one better off [51]. The importance of market equilibrium is evidenced by the 1972 Nobel Prize awarded to John R. Hicks and Kenneth J. Arrow "for their pioneering contributions to general economic equilibrium theory and welfare theory" [58]. Market equilibrium has wide application in fair allocation [32], as a few examples, fairly assigning course seats to students [11] or dividing estates, rent, fares, and others [35]. Besides, market equilibrium are also considered for ad auctions with budget constraints where money has real value [15, 16].

Existing works often use traditional optimization method or online learning technique to solve market equilibrium, which can tackle one market with around 400 buyers and goods in experiments [30, 52]. However, in realistic scenarios, there might be millions of buyers in one market (*e.g.* job market, online shopping market). In these scenarios, the description complexity for the market is $O(nm)$ and it needs at least $O(nm)$ cost to do one optimization step for the market, if there are $n$ buyers and $m$ goods in the market, which is unacceptable when $n$ is extremely large and potentially infinite. In this case, and traditional optimization methods do not work anymore.

However, contextual models come to the rescue. The success of contextual auctions[21, 5] demonstrate the power of contextual models, in which each bidder and item are represented as context and

the value (or the distribution) of item to bidder is determined by the contexts. In this way, auctions as well as other economic problems can be described in a more memory-efficient way, making it possible to accelerate the computation on these problems. Inspired by the models of contextual auctions, we propose the concept of contextual markets in a similar way. We verify that contextual markets can be useful to model large-scale markets aforementioned, since the real market can be assumed to be within some low dimension space, and the values of goods to buyers are often not hard to speculate given the knowledge of goods and buyers [46, 45]. Besides, contextual models never lose expressive power compared with raw models[7], giving contextual markets capabilities to generalize over traditional markets.

This paper initiates the study of *deep learning* for *contextual* market equilibrium computation with a large number of buyers. The description complexity of contextual markets is $O(n + m)$, if there are $n$ buyers and $m$ items in the market, making them memory-efficient and helpful for follow-up equilibrium computation while holding the market structure. Following the framework of differentiable economics [18, 26, 62], we propose a deep-learning based approach, MarketFCNet, in which one optimization step costs only $O(m)$ rather than $O(nm)$ in traditional methods, greatly accelerating the computation of market equilibrium. MarketFCNet takes the representations of one buyer and one good as input, and outputs the allocation of the good to the buyer. The training on MarketFCNet targets at an unbiased estimator of the objective function of EG-convex program, which can be formed by independent samples of buyers. By this way, we optimize the allocation function on "buyer space" implicitly, rather than optimizing the allocation to each buyer directly. Therefore, MarketFCNet can reduce the algorithm complexity such that it becomes independent of $n$, *i.e.*, the number of buyers.

The effectiveness of MarketFCNet is demonstrated by our experimental results. As the market scale expands, MarketFCNet delivers competitive performance and significantly lower running times compared to existing methods in different experimental settings, demonstrating the potential of deep learning-based methods to accelerate the approximation of large-scale contextual market equilibrium.

The contributions of this paper consist of three parts,

- We proposes a method, MarketFCNet, to approximate the contextual market equilibrium in which the number of buyers is large.

- We proposes Nash Gap to quantify the deviation of the given allocation and price pair from the market equilibrium.

- We conduct extensive experiments, demonstrating promising performance on the approximation measure and running time compared with existing methods.

## 2 Related Works

The history of market equilibrium arises from microeconomics theory, where the concept of competitive equilibrium [51, §10] was proposed, and the existence of market equilibrium is guaranteed in a general setting [3, 61]. Eisenberg and Gale [28] first considered the linear market case, and proved that the solution of EG-convex program constitutes a market equilibrium, which lays the polynomial-time algorithmic foundations for market equilibrium computation. Eisenberg [27] later showed that EG program also works for a class of CCNH utility functions. Shmyrev program later is also proposed to solve market equilibrium with linear utility with a perspective shift from allocation to price [57], while Cole et al. [14] later found that Shmyrev program is the dual problem of EG program with a change of variables. There are also a branch of literature that consider computational perspective in more general settings such as indivisible goods [54, 19, 20] and piece-wise linear utility [60, 33, 34].

There are abundant of works that present algorithms to solve the market equilibrium and shows the convergence results theoretically [13]. Gao and Kroer [30] discusses the convergence rates of first-order algorithms for EG convex program under linear, quasi-linear and Leontief utilities. Nan et al. [52] later designs stochastic optimization algorithms for EG convex program and Shmyrev program with convergence guarantee and show some economic insight. Jalota et al. [42] proposes an ADMM algorithm for CCNH utilities and shows linear convergence results. Besides, researchers are more engaged in designing dynamics that possess more economic insight. For example, PACE

dynamic [32, 48, 65] and proportional response dynamic [63, 66, 12], though the original idea of PACE arise from auction design [16, 15].

With the fast growth of machine learning and neural network, many existing works aim at resolving economic problem by deep learning approach, which falls into the differentiate economy framework [26]. A mainstream is to approximate the optimal auction with differentiable models by neural networks [25, 29, 36, 55]. The problem of Nash equilibrium computation in normal form games [22, 50, 23] and optimal contract design [62] through deep learning also attracts researchers' attentions. Among these methodologies, transformer architecture [50, 21, 47] is widely used in solving economic problems.

To the best of our knowledge, no existing works try to approximate market equilibrium through deep learning. Besides, although some literature focuses on low-rank markets and representative markets [46, 45], our works firstly propose the concept of contextual market. We believe that our approach will pioneer a promising direction for large-scale contextual market equilibrium computation.

## 3 Contextual Market Modelling

In this section, we focus on the model of contextual market equilibrium in which goods are assumed to be divisible. Let the market consist of $n$ buyers, denoted as $1, ..., n$, and $m$ goods, denoted as $1, ..., m$. We denote $[k]$ as the abbreviation of the set $\{1, 2, \ldots, k\}$. Each buyer $i \in [n]$ has a representation $b_i$, and each good $j \in [m]$ has a representation $g_j$. We assume that $b_i$ belongs to the buyer representation space $\mathcal{B}$, and $g_j$ belongs to the good representation space $\mathcal{G}$. For a buyer with representation $b \in \mathcal{B}$, she has budget $B(b) > 0$. Denote $Y(g) > 0$ as the supply of good with representation $g$. Although many existing works [30] assume that each good $j$ has *unit* supply (i.e. $Y(g) \equiv 1$ for all $g \in \mathcal{G}$) without loss of generality, their results can be easily generalized to our settings.

An *allocation* is a matrix $\boldsymbol{x} = (x_{ij})_{i \in [n], j \in [m]} \in \mathbb{R}_+^{n \times m}$, where $x_{ij}$ is the amount of good $j$ allocated to buyer $i$. We denote $\boldsymbol{x}_i = (x_{i1}, \ldots, x_{im})$ as the vector of bundle of goods that is allocated to buyer $i$. The buyers' utility function is denoted as $u : \mathcal{B} \times \mathbb{R}_+^m \to \mathbb{R}_+$, here $u(b_i; \boldsymbol{x}_i)$ denotes the utility of buyer $i$ with representation $b_i$ when she chooses to buy $\boldsymbol{x}_i$. We denote $u_i(\boldsymbol{x}_i)$ as an equivalent form of $u(b_i; \boldsymbol{x}_i)$ and often refer them as the same thing. Similarly, $B(b_i), Y(g_j)$ and $B_i, Y_j$ are often referred to as the same thing, respectively.

Let $\boldsymbol{p} = (p_1, \ldots, p_m) \in \mathbb{R}_+^m$ be the prices of the goods, the *demand set* of buyer with representation $b_i$ is defined as the set of utility-maximizing allocations within budget constraint.

$$D(b_i; \boldsymbol{p}) \coloneqq \arg\max_{\boldsymbol{x}_i} \left\{ u(b_i; \boldsymbol{x}_i) \mid \boldsymbol{x}_i \in \mathbb{R}_+^m, \langle \boldsymbol{p}, \boldsymbol{x}_i \rangle \leq B(b_i) \right\}. \tag{1}$$

A *contextual market* is a 4-tuple: $\mathcal{M} = \langle n, m, (b_i)_{i \in [n]}, (g_j)_{j \in [m]} \rangle$, where buyer utility $u(b_i; \boldsymbol{x}_i)$ is known given the information of the market. We also assume budget function $B : \mathcal{B} \to \mathbb{R}_+$ represents the budget of buyers and capacity function $Y : \mathcal{G} \to \mathbb{R}_+$ represents the supply of goods. All of $u, B$ and $Y$ are assumed to be public knowledge and excluded from a market representation. This assumption mainly comes from two aspects: (1) these functions can be learned from historical data and (2) budgets and supplies can be either encoded in $b$ and $g$ in some way.

The *market equilibrium* is represented as a pair $(\boldsymbol{x}, \boldsymbol{p})$, $\boldsymbol{x} \in \mathbb{R}_+^{n \times m}$, $\boldsymbol{p} \in \mathbb{R}_+^m$, which satisfies the following conditions.

- *Buyer optimality*: $\boldsymbol{x}_i \in D(b_i, \boldsymbol{p})$ for all $i \in [n]$,
- *Market clearance*: $\sum_{i=1}^n x_{ij} \leq Y(g_j)$ for all $j \in [m]$, and equality must hold if $p_j > 0$.

We say that $u_i$ is *homogeneous* (with degree 1) if it satisfies $u_i(\alpha \boldsymbol{x}_i) = \alpha u_i(\boldsymbol{x}_i)$ for any $\boldsymbol{x}_i \geq 0$ and $\alpha > 0$ [53, §6.2]. Following existing works, we assume that $u_i$s are CCNH utilities, where CCNH represents for concave, continuous, non-negative, and homogeneous functions[30]. For CCNH utilities, a market equilibrium can be computed using the following *Eisenberg-Gale convex program* (EG):

$$\max \sum_{i=1}^n B_i \log u_i(\boldsymbol{x}_i) \quad \text{s.t.} \sum_{i=1}^n x_{ij} \leq Y_j, \ \boldsymbol{x} \geq 0. \tag{EG}$$

Theorem 3.1 shows that the market equilibrium can be represented as the optimal solution of (EG).

**Theorem 3.1** (Gao and Kroer [30]). *Let $u_i$ be concave, continuous, non-negative and homogeneous (CCNH). Assume $u_i(\mathbf{1}) > 0$ for all $i$. Then, (i) (EG) has an optimal solution and (ii) any optimal solution $\boldsymbol{x}$ to (EG) together with its optimal Lagrangian multipliers $\boldsymbol{p}^* \in \mathbb{R}_+^m$ constitute a market equilibrium, up to arbitrary assignment of zero-price items. Furthermore, $\langle \boldsymbol{p}^*, \boldsymbol{x}_i^* \rangle = B_i$ for all $i$.*

Based on Theorem 3.1, it's easy to find that we can always assume $\sum_{i \in [n]} x_{ij} = Y_j$ while preserving the existence of market equilibrium, which states as follows.

**Proposition 3.2.** *Following the assumptions in Theorem 3.1. For the following EG convex program with equality constraints,*

$$\max \sum_{i=1}^n B_i \log u_i(\boldsymbol{x}_i) \quad \text{s.t.} \sum_{i=1}^n x_{ij} = Y_j, \ \boldsymbol{x} \geq 0. \tag{2}$$

*Then, an optimal solution $\boldsymbol{x}^*$ together with its Lagrangian multipliers $\boldsymbol{p}^* \in \mathbb{R}_+^m$ constitute a market equilibrium. Moreover, assume more that for each good $j$, there is some buyer $i$ such that $\frac{\partial u_i}{\partial x_{ij}} > 0$ always hold whenever $u_i(\boldsymbol{x}_i) > 0$, then all prices are strictly positive in market equilibrium. As a consequence, Equation (EG) and Equation (2) derive the same solution.*

We leave all proofs to Appendix B. Since the additional assumption in Proposition 3.2 is fairly weak, without further clarification, we always assume the conditions in Proposition 3.2 hold and the market clearance condition becomes $\sum_{i \in [n]} x_{ij} = Y(g_j), \ \forall j \in [m]$.

# 4 MarketFCNet

In this section, we introduce the MarketFCNet (denoted as Market Fully-Connected Network) approach to solve the market equilibrium when the number of buyers is large and potentially infinite. MarketFCNet is a sampling-based methodology, and the key point is to design an unbiased estimator of an objective function whose solution coincides with the market equilibrium. The main advantage is that it has the potential to fit the infinite-buyer case without scaling the computational complexity. Therefore, MarketFCNet is scalable with the number of buyers varies.

## 4.1 Problem Reformulation

Following the idea of differentiable economics [26], we consider parameterized models to represent the allocation of good $j$ to buyer $i$, denoted as $x_\theta(b_i, g_j)$, and call it allocation network, where $\theta$ is the network parameter. Given buyer $i$ and good $j$, the network can automatically compute the allocation $x_{ij} = x_\theta(b_i, g_j)$. The allocation to buyer $i$ is represented as $\boldsymbol{x}_i = \boldsymbol{x}_\theta(b_i, \boldsymbol{g})$ and the allocation matrix is represented as $\boldsymbol{x} = \boldsymbol{x}_\theta(\boldsymbol{b}, \boldsymbol{g})$. Then the market clearance constraint can be reformulated as $\sum_{i \in [n]} x_\theta(b_i, g_j) = Y(g_j), \forall j \in [m]$ and the price constraint can be reformulated as $\boldsymbol{x}_\theta(\boldsymbol{b}, \boldsymbol{g}) \geq 0$. Let $b$ be uniformly distributed from $\mathcal{B} = \{b_i : i \in [n]\}$, then the EG program (EG) becomes,

$$\begin{aligned} \max_{x_\theta} \quad & \text{OBJ}(x_\theta) = \mathbb{E}_b[B(b) \log u(b; \boldsymbol{x}_\theta(b, \boldsymbol{g}))] \\ \text{s.t.} \quad & \mathbb{E}_b[x_\theta(b, g_j)] = Y(g_j)/n, \forall j \in [m] \\ & \boldsymbol{x}_\theta(\boldsymbol{b}, \boldsymbol{g}) \geq 0 \end{aligned} \tag{EG-FC}$$

For simplicity, we take $Y(g_j)/n \equiv 1$ for all $g_j$.

## 4.2 Optimization

The second constraint in (EG-FC) can be easily handled by the network architecture (for example, network with a softplus layer $\sigma(x) = \log(1 + \exp(x))$. As for the first constraint, from Theorem 3.1, we know the prices of goods are simply the Lagrangian multipliers for the first constraint in (EG-FC). Therefore, we employ the Augmented Lagrange Multiplier Method (ALMM) to solve the problem (EG-FC). We define $\mathcal{L}_\rho(x_\theta, \lambda)$ as the Lagrangian, which has the form:

$$\mathcal{L}_\rho(x_\theta; \boldsymbol{\lambda}) = -\text{OBJ}(x_\theta) + \sum_{j=1}^m \lambda_j \left( \mathbb{E}_b[x_\theta(b, g_j)] - 1 \right) + \frac{\rho}{2} \sum_{j=1}^m \left( \mathbb{E}_b[x_\theta(b, g_j)] - 1 \right)^2 \tag{3}$$

Figure 1: Training process of MarketFCNet. On each iteration, the batch of $M$ independent buyers are drawn. each buyer and each good are represented as $k$-dimension context. The $(i, j)$'th element in the allocation matrix represents the allocation computed from $i$'th buyer and $j$'th good. MarketFCNet training process alternates between the training of allocation network and prices. The training of allocation network need to achieve an unbiased estimator $\widehat{\mathcal{L}}_\rho(x_\theta; \lambda)$ of the loss function $\mathcal{L}_\rho(x_\theta; \lambda)$, followed by gradient descent. The training of prices need to get an unbiased estimator $\widehat{\Delta}\lambda_j$ of $\Delta\lambda_j$, followed by ALMM updating rule $\lambda_j \leftarrow \lambda_j + \beta_t \widehat{\Delta}\lambda_j$.

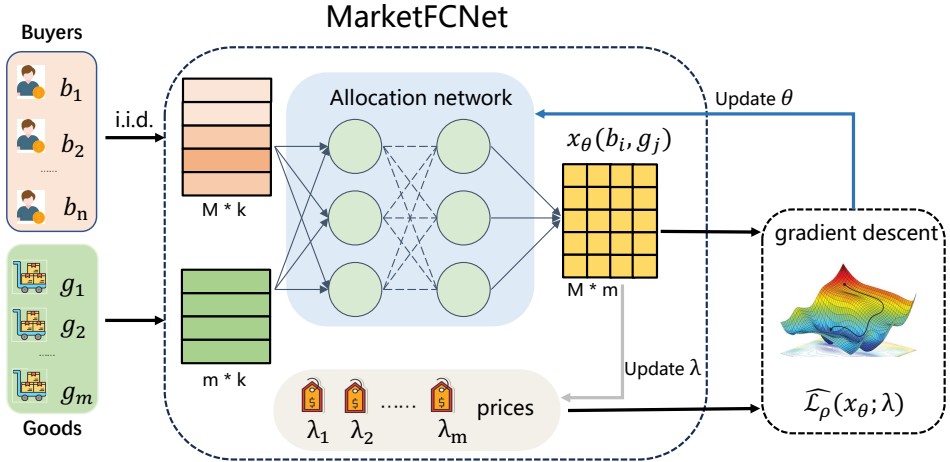

Directly computing the objective function seems intractable due to the potentially infinite data size. Therefore, we follow the framework in learning theory culture that we only guarantee to achieve an unbiased gradient of the objective function [1, 8]. The training process of MarketFCNet is presented in Figure 1.

To finish the ALMM algorithm, we need to obtain unbiased estimators of following two expressions.

- An unbiased estimator of $\mathcal{L}_\rho(x_\theta; \boldsymbol{\lambda})$.
- An unbiased estimator of $\Delta\lambda_j$, where $\Delta\lambda_j$ is given by $\Delta\lambda_j = \rho \left( \mathbb{E}_b[x_\theta(b, g_j)] - 1 \right)$.

**Unbiased estimator of $\Delta\lambda_j$** We aim to obtain an unbiased estimator of $\mathbb{E}_b[x_\theta(b, g_j)]$. By applying Monte Carlo method, we can choose batch size $M$ and sample $b_1, b_2, ..., b_M \sim U(\mathcal{B})$, then $\frac{1}{M} \sum_{i=1}^{M} x_\theta(b_i, g_j)$ forms an unbiased estimator.

**Unbiased estimator of $\mathcal{L}_p(x_\theta; \boldsymbol{\lambda})$** For $\text{OBJ}(x_\theta)$ and the second term, the technique to achieve an unbiased estimator is similar. $u(b; x_\theta(b, \boldsymbol{g}))$ in $\text{OBJ}(x_\theta)$ can be calculated directly by summing over all goods. For the last term, notice that

$$\left( \mathbb{E}_b \left[ x_\theta(b, g_j) \right] - 1 \right)^2 = \left( \mathbb{E}_b \left[ x_\theta(b, g_j) \right] - 1 \right) \cdot \left( \mathbb{E}_{b'} \left[ x_\theta(b', g_j) \right] - 1 \right) \tag{4}$$

Therefore, we can sample $b_1, ..., b_M, b'_1, ..., b'_M \sim U(\mathcal{B})$ and compute

$$\frac{\rho}{2} \cdot \frac{1}{M} \sum_{i=1}^{M} \sum_{j=1}^{m} \left( x_\theta(b_i, g_j) - 1 \right) \cdot \left( x_\theta(b'_i, g_j) - 1 \right) \tag{5}$$

which provides an unbiased estimator for the last term, capturing the squared deviation of output allocations from the constraint.

## 5 Performance Measures of Market Equilibrium

In this section, we propose *Nash Gap* to measure the performance of an approximated market equilibrium and show that Nash Gap preserves the economic interpretation for market equilibrium. To introduce Nash Gap, we first introduce two types of welfare, Log Nash Welfare and Log Fixed-price Welfare in Definition 5.1 and Definition 5.2, respectively.

**Definition 5.1** (Log Nash Welfare)**.** The Log Nash Welfare (abbreviated as LNW) is defined as

$$\text{LNW}(\boldsymbol{x}) = \frac{1}{B_{\text{total}}} \sum_{i \in [n]} B_i \log u_i(\boldsymbol{x}_i), \tag{6}$$

where $B_{\text{total}} = \sum_{i \in [n]} B_i$ is the total budgets for buyers.

Notice that $\text{LNW}(\boldsymbol{x})$ is identical to the objective function in Equation (EG), differing only in the constant term coefficient.

**Definition 5.2** (Fixed-price and Log Fixed-price Welfare)**.** We define the fixed-price utility for buyer $i$ as,

$$\tilde{u}(b_i; \boldsymbol{p}) = \max_{\boldsymbol{x}_i} \{ u(b_i; \boldsymbol{x}_i) \mid \boldsymbol{x}_i \in \mathbb{R}_+^m, \langle \boldsymbol{p}, \boldsymbol{x}_i \rangle \leq B(b_i) \} \tag{7}$$

which represents the optimal utility that buyer $i$ can obtain at the price level $\boldsymbol{p}$, regardless of the market clearance constraints. The Log Fixed-price Welfare (abbreviated as LFW) is defined as the logarithm of Fixed-price Welfare,

$$\text{LFW}(\boldsymbol{p}) = \frac{1}{B_{\text{total}}} \sum_{i \in [n]} B_i \log \tilde{u}_i(\boldsymbol{p}) \tag{8}$$

Based on these definitions, we present the definition of Nash Gap.

**Definition 5.3** (Nash Gap)**.** We define Nash Gap (abbreviated as NG) as the difference of Log Nash Welfare and Log Fixed-price Welfare, *i.e.*

$$\text{NG}(\boldsymbol{x}, \boldsymbol{p}) = \text{LFW}(\boldsymbol{p}) - \text{LNW}(\boldsymbol{x}) \tag{9}$$

## 5.1 Properties of Nash Gap

To show why NG is useful in the measure of market equilibrium, we first observe that,

**Proposition 5.4** (Price constraints)**.** *If $(\boldsymbol{x}, \boldsymbol{p})$ constitute a market equilibrium, the following identity always hold,*

$$\sum_{j \in [m]} p_j Y_j = \sum_{i \in [n]} B_i \tag{10}$$

Below, we state the most important theorem in this paper.

**Theorem 5.5.** *Let $(\boldsymbol{x}, \boldsymbol{p})$ be a pair of allocation and price. Assuming the allocation satisfies market clearance and the price meets price constraint, then we have $\text{NG}(\boldsymbol{x}, \boldsymbol{p}) \geq 0$.*

*Moreover, $\text{NG}(\boldsymbol{x}, \boldsymbol{p}) = 0$ if and only if $(\boldsymbol{x}, \boldsymbol{p})$ is a market equilibrium.*

Theorem 5.5 show that Nash Gap is an ideal measure of the solution concept of market equilibrium, since it holds following properties,

- $\text{NG}(\boldsymbol{x}, \boldsymbol{p})$ is continuous on the inputs $(\boldsymbol{x}, \boldsymbol{p})$.
- $\text{NG}(\boldsymbol{x}, \boldsymbol{p}) \geq 0$ always hold. (under conditions in Theorem 5.5)
- $\text{NG}(\boldsymbol{x}, \boldsymbol{p}) = 0$ if and only if $(\boldsymbol{x}, \boldsymbol{p})$ meets the solution concept.
- The computation of NG does not require the knowledge of an equilibrium point $(\boldsymbol{x}^*, \boldsymbol{p}^*)$

Since some may argue that $\text{NG}(\boldsymbol{x}, \boldsymbol{p})$ is not intuitive to understand, we consider some more intuitive measures, the Euclidean distance to the market equilibrium, *i.e.*, $||\boldsymbol{x} - \boldsymbol{x}^*||$ and $||\boldsymbol{p} - \boldsymbol{p}^*||$, as well as the difference on Weighted Social Welfare, $|\text{WSW}(\boldsymbol{x}) - \text{WSW}(\boldsymbol{x}^*)|$, where $\text{WSW}(\boldsymbol{x}) := \sum_{i \in [n]} \frac{B_i}{B_{\text{total}}} u_i(\boldsymbol{x}_i)$, and show the connection between NG and these intuitive measures.

**Proposition 5.6.** *Under some technical assumptions (which is presented in Appendix B.4), if $\text{NG}(\boldsymbol{x}, \boldsymbol{p}) = \varepsilon$, we have:*

- $||\boldsymbol{p} - \boldsymbol{p}^*|| = O(\sqrt{\varepsilon})$.

226     • $||\boldsymbol{x}_i - \boldsymbol{x}_i^*|| = O(\sqrt{\varepsilon})$ *for all i.*

227     • $|\mathrm{WSW}(\boldsymbol{x}) - \mathrm{WSW}(\boldsymbol{x}^*)| = O(\varepsilon).$

228 Finally, we give a saddle-point explaination for Nash Gap.

229 **Corollary 5.7.** *Within market clearance and price constraint, we have*

$$\min_{\boldsymbol{p}} \mathrm{LFW}(\boldsymbol{p}) = \max_{\boldsymbol{x}} \mathrm{LNW}(\boldsymbol{x}) \tag{11}$$

230 Corollary 5.7 provides an economic interpretation for GAP. Market equilibrium can be seen as the
231 saddle point over social welfare, and the social welfare for $\boldsymbol{x}$ can be actually implemented while
232 the social welfare for $\boldsymbol{p}$ is virtual and desired by buyers. Nash Gap measures the gap between the
233 "desired welfare" and the "implemented welfare" for buyers.

## 5.2   Measures in General Cases

235 Since NG only works for $(\boldsymbol{x}, \boldsymbol{p})$ that satisfies market clearance and price constraints, we generalize
236 the measure of NG to a more general case, which need to give a measure for all positive $(\boldsymbol{x}, \boldsymbol{p})$.

237 We first notice that any equilibrium must satisfy the conditions of *market clearance* and *price*
238 *constraint*, we first make a projection on arbitrary positive $(\boldsymbol{x}, \boldsymbol{p})$ to the space where these constraints
239 hold. Specifically, if we let

$$\alpha_j = \frac{V_j}{\sum_i x_{ij}}, \quad \tilde{x}_{ij} = x_{ij} \cdot \alpha_j \qquad\qquad \beta = \frac{\sum_i B_i}{\sum_j V_j p_j}, \quad \tilde{p}_j = \beta \cdot p_j \tag{12}$$

240 then $(\tilde{\boldsymbol{x}}, \tilde{\boldsymbol{p}})$ satisfies these constraints and we consider $\mathrm{NG}(\tilde{\boldsymbol{x}}, \tilde{\boldsymbol{p}})$ as the equilibrium measure.

241 Besides, we also need to measure how far is the point $(\boldsymbol{x}, \boldsymbol{p})$ to the space within the conditions of
242 *market clearance* and *price constraint.* we propose following two measurement, called Violation of
243 Allocation (abbreviated as VoA) and Violation of Price (abbreviated as VoP), respectively.

$$\mathrm{VoA}(\boldsymbol{x}) := \frac{1}{m} \sum_j |\log \alpha_j|, \qquad \mathrm{VoP}(\boldsymbol{p}) := |\log \beta| \tag{13}$$

244 From the expressions of VoA and VoP, we know that these two constraints hold if and only if
245 $\mathrm{VoA}(\boldsymbol{x}) = 0$ and $\mathrm{VoP}(\boldsymbol{p}) = 0$.

246 We argue that this projection is of economic meaning. If $(\boldsymbol{x}, \boldsymbol{p})$ constitute a market equilibrium
247 and we scale budget with a factor of $\beta$, then $(\boldsymbol{x}, \beta\boldsymbol{p})$ constitute a market equilibrium in the new
248 market. Similarly, if we scale the value for each buyer with factor $1/\boldsymbol{\alpha}$ (here $\boldsymbol{\alpha}$ can be a vector in
249 $\mathbb{R}_+^m$) and capacity with factor $\alpha$, then, $(\boldsymbol{\alpha}\boldsymbol{x}, \frac{1}{\boldsymbol{\alpha}}\boldsymbol{p})$ constitute a market equilibrium in the new market.
250 These instances are evidence that market equilibrium holds a linear structure over market parameters.
251 Therefore, a linear projection can eliminate the effect from linear scaling, while preserving the effect
252 from orthogonal errors.

253 Notice that $\boldsymbol{x} = \tilde{\boldsymbol{x}}$ and $\boldsymbol{p} = \tilde{\boldsymbol{p}}$ if and only if $\mathrm{VoA}(\boldsymbol{x}) = 0$ and $\mathrm{VoP}(\boldsymbol{p}) = 0$, respectively. From
254 Theorem 5.5 We can easy derive following statements:

255 **Proposition 5.8.** *For arbitrary* $\boldsymbol{x} \in \mathbb{R}_+^{n \times m}, \boldsymbol{p} \in \mathbb{R}_+^m$, *we have* $\mathrm{VoA}(\boldsymbol{x}) \geq 0, \mathrm{VoP}(\boldsymbol{p}) \geq$
256 $0, \mathrm{NG}(\tilde{\boldsymbol{x}}, \tilde{\boldsymbol{p}}) \geq 0$ *always hold. Moreover,* $(\boldsymbol{x}, \boldsymbol{p})$ *is a market equilibrium if and only if* $\mathrm{VoA}(\boldsymbol{x}) =$
257 $\mathrm{VoP}(\boldsymbol{p}) = \mathrm{NG}(\tilde{\boldsymbol{x}}, \tilde{\boldsymbol{p}}) = 0.$

258 Proposition 5.8 is a certificate that $\mathrm{VoA}(\boldsymbol{x}), \mathrm{VoP}(\boldsymbol{p}), \mathrm{NG}(\tilde{\boldsymbol{x}}, \tilde{\boldsymbol{p}})$ together form a good measure for
259 market equilibrium. Therefore, in our experiments we compute these measures of solutions and
260 prefer a lower measure without further clarification.

## 6   Experiments

262 In this section, we present empirical experiments that evaluate the effectiveness of MarketFCNet.
263 Though briefly mentioned in this section, we leave the details of baselines, implementations, hyper-
264 parameters and experimental environments to Appendix C.

Table 1: Comparison of MarketFCNet with baselines: $n = 1,048,576$ buyers and $m = 10$ goods. The GPU time for MarketFCNet represents the training time and testing time, respectively.

| Methods | NG | VoA | VoP | GPU Time |
|---------|-----|-----|-----|----------|
| Naïve | 3.65e-1 | 0 | 0 | 3.57e-3 |
| EG | 2.17e-2 | 2.620e-1 | 7.031e-2 | 197 |
| EG-m | **2.49e-4** | 6.01e-2 | 9.77e-2 | 100 |
| FC | 1.63e-3 | **1.416e-2** | **6.750e-3** | **43.6**; **9.63e-2** |

## 6.1 Experimental Settings

In our experiments, all utilities are chosen as CES utilities, which captures a wide utility class including linear utilities, Cobb-Douglas utilities and Leontief utilities and has been widely studied in literature [59, 4]. CES utilities have the form,

$$u_i(x_i) = \left( \sum_{j \in [m]} v_{ij}^\alpha x_{ij}^\alpha \right)^{1/\alpha}$$

with $\alpha \leq 1$. The fixed-price utilities for CES utility is derived in Appendix A.

In order to evaluate the performance of MarketFCNet, we compare them mainly with a baseline that directly maximizes the objective in EG convex program with gradient ascent algorithm (abbreviated as *EG*), which is widely used in the field of market equilibrium computation. Besides, we also consider a momentum version of *EG* algorithm with momentum $\beta = 0.9$ (abbreviated as *EG-m*). We move the details of all baselines, experimental environments and implementations of algorithms to Appendix C.1 and Appendix C.2.

We also consider a naïve allocation and pricing rule (abbreviated as *Naïve*), which can be regarded as the benchmark of the experiments:

$$x_{ij} = 1, \quad p_j = \frac{\sum_{i \in [n]} B_i}{m V_j}, \quad \text{for all } i, j \tag{14}$$

In the following experiments, MarketFCNet is abbreviated as *FC*. Notice that *Naïve* always gives an allocation that satisfies market clearance and price constraints, while *EG*, *EG-m* and *FC* do not.

## 6.2 Experiment Results

**Comparing with Baselines**   We choose number of buyers $n = 1,048,576 = 2^{20}$, number of items $m = 10$, CES utilities parameter $\alpha = 0.5$ and representation with standard normal distribution as the basic experimental environment of MarketFCNet; We consider $\text{NG}(\tilde{x}, \tilde{p})$, $\text{VoA}(x)$, $\text{VoP}(p)$ and the running time of algorithms as the measures. Without special specification, these parameters are default settings among other experiments. Results are presented in Table 1. From these results we can see that the approximations of MarketFCNet are competitive with *EG* and *EG-m* and far better than Naïve, which means that the solution of MarketFCNet are very close to market equilibrium. MarketFCNet also achieve a much lower running time compared with *EG* and *EG-m*, which indicates that these methods are more suitable to large-scale market equilibrium computation. In following experiments, VoA and VoP measures are omitted and we only report NG and running time.

**Experiments in different parameters settings**   In this experiments, the market scale is chosen as $n = 4,194,304$ and $m = 10$. We consider experiments on different distribution of representation, including normal distribution, uniform distribution and exponential distribution. See (a) and (b) in Figure 2 for results. We also consider different $\alpha$ in our experimental settings. Specifically, our settings consist of: 1) $\alpha = 1$, the utility functions are linear; 2) $\alpha = 0.5$, where goods are substitutes; 3) $\alpha = 0$, where goods are neither substitutes or complements; 4) $\alpha = -1$, where goods are complements. More detailed results are shown in (c) and (d) Figure 2. The performance of MarketFCNet is robust in both settings.

Figure 2: The Nash Gap and GPU running time for different algorithms: MarketFCNet, EG and EG-m. Different colors represent for different algorithm. Market size is chosen as $n = 4, 194, 304$ buyers and $m = 10$ goods.

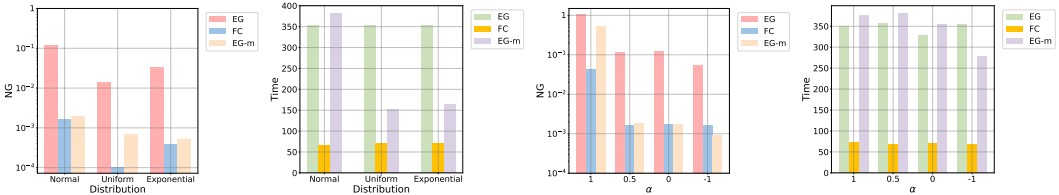

(a) Nash Gap on different context distributions.

(b) GPU running time on different context distributions.

(c) Nash Gap on different CES utilities parameter $\alpha$.

(d) GPU running time on different CES utilities parameter $\alpha$.

Figure 3: The Nash Gap and GPU running time for different algorithms: MarketFCNet, EG and EG-m. Different colors represent for different algorithm. Market size is chosen as $n = 2^{18}, 2^{20}, 2^{22}$ buyers and $m = 5, 10, 20$ goods.

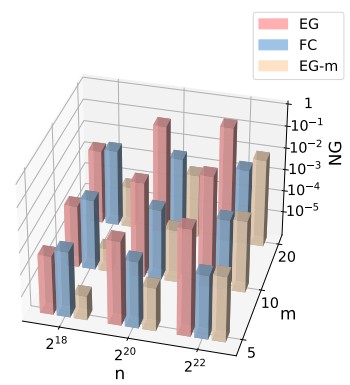

(a) Nash Gap on different market size, $n = 2^{18}, 2^{20}, 2^{22}$ buyers and $m = 5, 10, 20$ goods.

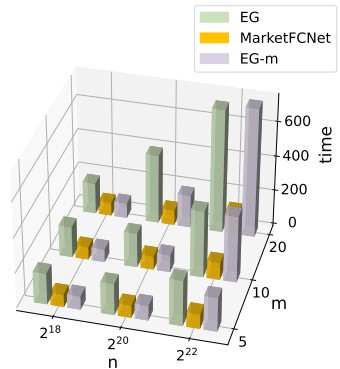

(b) GPU running time on different market size, $n = 2^{18}, 2^{20}, 2^{22}$ buyers and $m = 5, 10, 20$ goods.

**Different market scale for MarketFCNet** In this section we ask that how market size (here $n$ and $m$) will have impact on the efficiency of MarketFCNet. We set $m = 5, 10, 20$ and $n = 2^{18} = 262, 114, 2^{20} = 1, 048, 576, 2^{22} = 4, 194, 304$ as the experimental settings. For each combination of $n$ and $m$, we train MarketFCNet and compared with EG and EG-m, see results in Figure 3. As the market size varies, MarketFCNet has almost the same Nash Gap and running time, which shows the robustness of MarketFCNet method over different market sizes. However, as the market size increases, both EG and EG-m have larger Nash Gaps and longer running times, demonstrating that MarketFCNet is more suitable to large-scale contextual market equilibrium computation.

## 7 Conclusions and Future Work

This paper initiates the problem of large-scale contextual market equilibrium computation from a deep learning perspective. We believe that our approach will pioneer a promising direction for large-scale contextual market equilibrium computation.

For future works, it would be promising to extend these methods to the case when only the number of goods is large, or both the numbers of goods and buyers are large, which stays a blank throughout our works. Since many existing works proposed dynamics for online market equilibrium computation, it's also promising to extend our approaches to tackle the online market setting with large buyers. Besides, both existing works and ours consider sure budgets and values for buyers, and it would be interesting to extend the fisher market and equilibrium concept when the budgets or values of buyers are stochastic or uncertain.

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

# Appendix

# A    Derivation of Fixed-price Utility for CES Utility Functions

In this section we show the explicit expressions of Fixed-price Utility for CES utility functions.

We first consider the case $\alpha \neq 0, 1, -\infty$. The optimization problem for consumer $i$ is:

$$\max_{x_{ij}, j \in [m]} \quad u_i(\boldsymbol{x}_i) = \left[ \sum_{j \in [m]} v_{ij}^{\alpha} x_{ij}^{\alpha} \right]^{1/\alpha} \tag{15}$$

$$s.t. \quad \sum_{j \in [m]} x_{ij} p_j = B_i \tag{Budget Constraint}$$

$$x_{ij} \geq 0 \tag{16}$$

Not hard to verify that in an optimal solution with Equation (Budget Constraint), Equation (16) always holds, therefore we omit this constraint in our derivation.

We write the Lagrangian $L(\boldsymbol{x}_i, \lambda)$

$$L(\boldsymbol{x}_i, \lambda) = u_i(\boldsymbol{x}_i) + \lambda(B_i - \sum_{j \in [m]} x_{ij} p_j) \tag{17}$$

By $\frac{\partial L}{\partial x_{ij}} = 0$, we have

$$\frac{\partial u_i}{\partial x_{ij}^*}(\boldsymbol{x}_i) = \lambda p_j \tag{18}$$

We derive that

$$\frac{\partial u_i}{\partial x_{ij}}(\boldsymbol{x}_i) = \frac{1}{\alpha} \left[ \sum_{j \in [m]} v_{ij}^{\alpha} x_{ij}^{\alpha} \right]^{1/\alpha - 1} \cdot \alpha v_{ij}^{\alpha} x_{ij}^{\alpha - 1} \tag{19}$$

$$v_{ij}^{\alpha} x_{ij}^{\alpha - 1} = c p_j \qquad \cdots \text{let } c = \lambda \cdot \left[ \sum_{j \in [m]} v_{ij}^{\alpha} x_{ij}^{\alpha} \right]^{1/\alpha - 1} \tag{20}$$

$$x_{ij}^* = \frac{v_{ij}^{\frac{\alpha}{1-\alpha}}}{c^{\frac{1}{1-\alpha}} \cdot p_j^{\frac{1}{1-\alpha}}} \tag{21}$$

Taking (21) into (Budget Constraint), we get

$$B_i = \sum_{j \in [m]} \frac{v_{ij}^{\frac{\alpha}{1-\alpha}}}{c^{\frac{1}{1-\alpha}}} \cdot p_j^{-\frac{\alpha}{1-\alpha}} \tag{22}$$

$$c^{\frac{1}{1-\alpha}} = \frac{1}{B_i} \sum_{j \in [m]} \left( \frac{v_{ij}}{p_j} \right)^{\frac{\alpha}{1-\alpha}} \tag{23}$$

Taking Equation (23) into Equation (21), we get

$$x_{ij}^* = \frac{v_{ij}^{\frac{\alpha}{1-\alpha}}}{p_j^{\frac{1}{1-\alpha}}} \cdot \frac{B_i}{c_0} \tag{24}$$

where $c_0 = \sum_{j \in [m]} \left( \frac{v_{ij}}{p_j} \right)^{\frac{\alpha}{1-\alpha}}$

Taking Equation (24) into Equation (15), we finally have

$$
\begin{aligned}
u_i(\boldsymbol{x}_i^*) &= \left[ v_{ij}^\alpha x_{ij}^\alpha \right]^{\frac{1}{\alpha}} \\
&= \left[ \sum_{j \in [m]} v_{ij}^\alpha \frac{v_{ij}^{\frac{\alpha^2}{1-\alpha}}}{p_j^{\frac{\alpha}{1-\alpha}}} c_0^\alpha \right] \\
&= \left[ \sum_{j \in [m]} \left( \frac{v_{ij}}{p_j} \right)^{\frac{\alpha}{1-\alpha}} c_0^\alpha \right] \\
&= B_i c_0^{\frac{1-\alpha}{\alpha}}
\end{aligned}
\tag{25}
$$

$$\log \tilde{u}_i(\boldsymbol{p}) = \log u_i(\boldsymbol{x}_i^*) = \log B_i + \frac{1-\alpha}{\alpha} \log c_0$$

For $\alpha = 1$, by simple arguments we know that consumer will only buy the good that with largest value-per-cost, i.e., $v_{ij}/p_j$. Therefore, we have

$$\log \tilde{u}_i(\boldsymbol{p}) = \log B_i + \log \max_j \frac{v_{ij}}{p_j} \tag{26}$$

For $\alpha = 0$, we have $\log u_i(\boldsymbol{x}_i) = \frac{1}{v_t} \sum_{j \in [m]} v_{ij} \log x_{ij}$ where $v_t = \sum_{j \in [m]} v_{ij}$.

Similarly, we have

$$cp_j = \frac{\partial \log u_i}{\partial x_{ij}} = \frac{v_{ij}}{x_{ij}} \tag{27}$$

$$x_{ij}^* = \frac{v_{ij}}{cp_j} \tag{28}$$

By solving budget constraints we have $c = \frac{v_t}{B_i}$, and therefore, $x_{ij}^* = \frac{v_{ij} B_i}{p_j v_t}$ and

$$\log u_i(\boldsymbol{x}_i^*) = \frac{1}{v_t} \sum_{j \in [m]} \left( v_{ij} \log \frac{v_{ij} B_i}{p_j v_t} \right) \tag{29}$$

$$= \log B_i + \sum_{j \in [m]} \frac{v_{ij}}{v_t} \log \frac{v_{ij}}{p_j v_t} \tag{30}$$

For $\alpha = -\infty$, we can easily know that $v_{ij} x_{ij}^* \equiv c$ for some $c$. By solving budget constraint we have

$$\sum_{j \in [m]} \frac{cp_j}{v_{ij}} = B_i \tag{31}$$

$$c = B_i \left( \sum_{j \in [m]} \frac{p_j}{v_{ij}} \right)^{-1} \tag{32}$$

$$\log \tilde{u}_i(\boldsymbol{p}) = \log c = \log B_i - \log \sum_{j \in [m]} \frac{p_j}{v_{ij}} \tag{33}$$

Above all, the log Fixed-price Utility for CES functions is

$$\log \tilde{u}_i(\boldsymbol{p}) = \begin{cases} \log B_i + \max_j \log \frac{v_{ij}}{p_j} & \text{for } \alpha = 1 \\ \log B_i + \sum_{j\in[m]} \frac{v_{ij}}{v_t} \log \frac{v_{ij}}{p_j v_t} & \text{for } \alpha = 0 \\ \log B_i - \log \sum_{j\in[m]} \frac{p_j}{v_{ij}} & \text{for } \alpha = -\infty \\ \log B_i + \frac{1-\alpha}{\alpha} \log c_0 & \text{others} \end{cases} \tag{34}$$

# B  Omitted Proofs

## B.1  Proof of Proposition 3.2

We consider Lagrangian multipliers $\boldsymbol{p}$ and use the KKT condition. The Lagrangian becomes

$$L(\boldsymbol{p}, \boldsymbol{x}) = \sum_i B_i \log u_i(\boldsymbol{x}_i) - \sum_j p_j (\sum_i x_{ij} - Y_j) \tag{35}$$

and the partial derivative of $x_{ij}$ is

$$\frac{\partial L(\boldsymbol{p}, \boldsymbol{x}_i)}{\partial x_{ij}} = \frac{B_i}{u_i(\boldsymbol{x}_i)} \frac{\partial u_i}{\partial x_{ij}} - p_j \tag{36}$$

By complementary slackness of $x_{ij} \geq 0$, we have

$$\frac{B_i}{u_i(\boldsymbol{x}_i)} \frac{\partial u_i}{\partial x_{ij}} \leq p_j \text{ for all } i \tag{37}$$

By theorem 3.1, we know that if $(\boldsymbol{x}, \boldsymbol{p})$ is a market equilibrium, we must have $u_i(\boldsymbol{x}_i) > 0$ for all $i$, and by condition in Proposition 3.2, we can always select buyer $i$ such that $\frac{\partial u_i}{\partial x_{ij}} > 0$. Therefore, we have $p_j > 0$.

As a consequence, $p_j > 0$ indicates that $\sum_j x_{ij} = V_j$ by market clearance condition.

## B.2  Proof of Proposition 5.4

Consider the market equilibrium condition $\langle \boldsymbol{p}^*, \boldsymbol{x}_i^* \rangle = B_i$, we have $\sum_j p_j x_{ij} = B_i$. sum over this expression, we have $\sum_i \sum_j p_j x_{ij} = \sum_i B_i$. Then, $\sum_j p_j \sum_i x_{ij} = \sum_i B_i$. Notice that we have $\sum_{i=1}^n x_{ij} = Y_j$ in market equilibrium, so $\sum_j p_j Y_j = \sum_i B_i$, that completes the proof.

## B.3  Proof of Theorem 5.5

*Proof of Theorem 5.5.* Denote $(\boldsymbol{x}, \boldsymbol{p})$ as the market equilibrium, $\boldsymbol{p}$ as the price for goods and $\boldsymbol{x}_i^*(\boldsymbol{p})$ as the optimal consumption set of buyer $i$ when the price is $\boldsymbol{p}$.

We have following equation:

$$\sum_j x_{ij} p_j = B_i \tag{38}$$

$$\boldsymbol{x}_i \in \boldsymbol{x}_i^*(\boldsymbol{p}) \tag{39}$$

$$\sum_{i\in[n]} x_{ij} = Y_j \tag{40}$$

$$u_i(\boldsymbol{p}) = u_i(\boldsymbol{x}_i), \ \forall \boldsymbol{p} \in \mathbb{R}_+^m, \ \forall \boldsymbol{x}_i \in \boldsymbol{x}_i^*(\boldsymbol{p}) \tag{41}$$

From Proposition 5.4 we know $\sum_{i\in[n]} B_i = \sum_{j\in[m]} Y_j p_j$.

Let $\boldsymbol{p}'$ be some price for items such that $\sum_{j\in[m]} Y_j p_j' = \sum_{i\in[n]} B_i$. Let $\boldsymbol{x}_i' \in \boldsymbol{x}_i^*(\boldsymbol{p}')$ and $B_i' = \langle \boldsymbol{p}', \boldsymbol{x}_i \rangle$. We know that

$$\sum_{i\in[n]} B_i' = \langle \boldsymbol{p}', \sum_{i\in[n]} \boldsymbol{x}_i \rangle = \langle \boldsymbol{p}', \boldsymbol{Y} \rangle = \sum_{i\in[n]} B_i \tag{42}$$

530 For consumer $i$, $\boldsymbol{x}_i$ costs $B_i'$ at price $\boldsymbol{p}'$, thus $\frac{B_i}{B_i'}\boldsymbol{x}_i$ costs $B_i$ at price $\boldsymbol{p}'$. Besides, $\boldsymbol{x}_i'$ also costs $B_i$ for
531 price $\boldsymbol{p}'$, and $\boldsymbol{x}'$ is the optimal consumption for buyer $i$. Then we have

$$u_i(\boldsymbol{p}') = u_i(\boldsymbol{x}_i') \geq u_i(\frac{B_i}{B_i'}\boldsymbol{x}_i) = \frac{B_i}{B_i'}u_i(\boldsymbol{x}_i) \tag{43}$$

532 where the last equation is from the homogeneity(with degree 1) of utility function.

533 Taking logarithm and weighted sum with $B_i$, we have

$$\sum_{i\in[n]} B_i \log u_i(\boldsymbol{p}') \geq \sum_{i\in[n]} B_i \log \frac{B_i}{B_i'} + \sum_{i\in[n]} B_i \log u_i(\boldsymbol{x}_i) \tag{44}$$

534 Take $B_{\text{total}} = \sum_{i\in[n]} B_i$, the first term in RHS becomes

$$\sum_{i\in[n]} B_i \log \frac{B_i}{B_i'} \tag{45}$$

$$=B_{\text{total}} \sum_{i\in[n]} \left( \frac{B_i}{B_{\text{total}}} \log \frac{B_i/B_{\text{total}}}{B_i'/B_{\text{total}}} \right) \tag{46}$$

$$=B_{\text{total}} \cdot \text{KL}(\frac{\boldsymbol{B}}{B_{\text{total}}} \| \frac{\boldsymbol{B}'}{B_{\text{total}}}) \tag{47}$$

$$\geq 0 \tag{48}$$

535 Therefore,

$$\sum_{i\in[n]} B_i \log u_i(\boldsymbol{p}') \geq \sum_{i\in[n]} B_i \log u_i(\boldsymbol{x}_i) \tag{49}$$

536 For $\boldsymbol{x}'$ that satisfies market clearance, by optimality of EG program(EG), we have

$$\sum_{i\in[n]} B_i \log u_i(\boldsymbol{x}_i) \geq \sum_{i\in[n]} B_i \log u_i(\boldsymbol{x}_i') \tag{50}$$

537 Equation (49) and Equation (50) together complete the proof of the first part.

538 If $(\boldsymbol{x},\boldsymbol{p})$ constitutes a market equilibrium, it's obvious that $\text{LFW}(\boldsymbol{p})$ and $\text{LNW}(\boldsymbol{x})$ are identical,
539 therefore $\text{NG}(\boldsymbol{x},\boldsymbol{p}) = 0$.

540 On the other hand, if $(\boldsymbol{x},\boldsymbol{p})$ is not a market equilibrium, but $\text{NG}(\boldsymbol{x},\boldsymbol{p}) = 0$, it means that the KL
541 convergence term must equal to 0, and $B_i = B_i'$ for all $i$, which means that $\boldsymbol{x}_i$ costs buyer $i$ with
542 money $B_i$ and $\boldsymbol{x}_i$ are in the consumption set of buyer $i$. Since $(\boldsymbol{x},\boldsymbol{p})$ is not a market equilibrium,
543 there is at least one buyer that can choose a better allocation $\boldsymbol{x}_i'$ to improve her utility, therefore
544 improve $\text{LFW}(\boldsymbol{p})$, and it cannot be the case that $\text{LFW}(\boldsymbol{p}) = \text{LNW}(\boldsymbol{x})$, which makes a contradiction.

545 $\square$

## B.4 Proof of Proposition 5.6

547 We leave the formal presentation of Proposition 5.6 and proofs to three theorems below.

548 **Lemma B.1.** *Assume that $u_i(\boldsymbol{x}_i)$ is twice differentiable and denote $H(\boldsymbol{x}_i)$ as the Hessian matrix of*
549 *$u_i(\boldsymbol{x}_i)$. If following hold:*

550      • *$H(\boldsymbol{x}_i^*)$ has rank $m-1$*

551      • *$\|\boldsymbol{x}_i - \boldsymbol{x}_i^*\| = \varepsilon$ for some $i$*

552      • *$\boldsymbol{x}_i^* > \boldsymbol{0}$*

553 *then we have $\text{OPT} - \text{LNW}(\boldsymbol{x}) = \Omega(\varepsilon^2)$.*

**Lemma B.2.** *Denote $\tilde{u}_i(\boldsymbol{p}, B_i)$ and $\boldsymbol{x}_i^*(\boldsymbol{p}, B_i)$ as the maximum utility buyer $i$ can get and the corresponding consumption for buyer $i$ when her budget is $B_i$ and prices are $\boldsymbol{p}$. If following hold:*

- $||\boldsymbol{p} - \boldsymbol{p}^*|| = \varepsilon$

- $\boldsymbol{x}_i^*(\boldsymbol{p}, B_i)$ *is differentiable with $\boldsymbol{p}$.*

- $H_X := (\sum_{i \in [n]} \frac{\partial x_{ij}^*}{\partial p_k}(\boldsymbol{p}^*, B_i))_{j,k \in [m]}$ *has full rank.*

*then we have $LFW(\boldsymbol{p}) - OPT = \Omega(\varepsilon^2)$.*

*Remark* B.3. It's worth notice that $H(\boldsymbol{x}_i^*)$ can not has full rank $m$, since $u_i(\boldsymbol{x})$ is assumed to be homogeneous and thus linear in the direction $\boldsymbol{x}$. Therefore, we have $H(\boldsymbol{x}_i)\boldsymbol{x}_i = \boldsymbol{0}$ for all $\boldsymbol{x}_i$.

Let $C_i = \{\boldsymbol{x}_i \in \mathbb{R}_+^m : \langle \boldsymbol{p}, \boldsymbol{x}_i \rangle = B_i\}$ be the consumption set of buyer $i$, since $\boldsymbol{x}_i$ can not be parallel with $C_i$, the condition that $H(\boldsymbol{x}_i^*)$ has rank $m-1$ means that, $H(\boldsymbol{x}_i)$ is strongly concave at point $\boldsymbol{x}_i^*$ on the consumption set $C_i$.

Besides, we emphasize that the conditions in Lemma B.1 and Lemma B.2 are satisfied for CES utility with $\alpha < 1$.

**Corollary B.4.** *Under the assumptions in Lemma B.1 and Lemma B.2, if $\mathrm{NG}(\boldsymbol{x}, \boldsymbol{p}) = \varepsilon$, we have:*

- $||\boldsymbol{p} - \boldsymbol{p}^*|| = O(\sqrt{\varepsilon})$

- $||\boldsymbol{x}_i - \boldsymbol{x}_i^*|| = O(\sqrt{\varepsilon})$ *for all $i$*

*Proof of Corollary B.4.* A direct inference from Lemma B.1 and Lemma B.2, notice that $\mathrm{NG} = \varepsilon$ indicates that $\mathrm{OPT} - \mathrm{LNW}(\boldsymbol{x}) \leq \varepsilon$ and $\mathrm{LFW}(\boldsymbol{p}) - \mathrm{OPT} \leq \varepsilon$. $\square$

Corollary B.4 states that, for a pair of $(\boldsymbol{x}, \boldsymbol{p})$ that satisfy market clearance and price constraints, a small Nash Gap indicates that the point $(\boldsymbol{x}, \boldsymbol{p})$ is close to the equilibrium point $(\boldsymbol{x}^*, \boldsymbol{p}^*)$, in the sense of Euclidean distance.

**Lemma B.5.** *Assume following hold:*

- *buyers have same utilities at $\boldsymbol{x}^*$, i.e. $u_i(\boldsymbol{x}_i^*) = u_j(\boldsymbol{x}_j^*) \equiv c$ for all $i$, $j$*

- $||\boldsymbol{x}_i - \boldsymbol{x}_i^*|| \leq \varepsilon$ *for all $i$*

*then, we have $|\mathrm{WSW}(\boldsymbol{x}) - \mathrm{WSW}(\boldsymbol{x}^*)| = O(\varepsilon^2)$.*

*Remark* B.6. These conditions can be held when buyers are homogeneous, *i.e.*, $B_i = B_j$ and $u_i(\boldsymbol{x}) = u_j(\boldsymbol{x})$ for all $i, j, \boldsymbol{x} \in \mathbb{R}_+^m$. Besides, consider buyers with same budgets, these conditions can also be held if the market has some "equivariance property", *e.g.*, there is a $n$-cycle permutation of buyers $\rho$ and permutation of goods $\tau$, such that $u_i(\boldsymbol{x}_i) = u_{\rho(i)}(\tau(\boldsymbol{x}_{\rho(i)}))$ for all $i$ and $\tau(Y_1, ..., Y_m) = (Y_1, ..., Y_m)$.

**Corollary B.7.** *Under the assumptions in Lemma B.1 and Lemma B.5, if $\mathrm{NG}(\boldsymbol{x}, \boldsymbol{p}) = \varepsilon$, we have*

- $|\mathrm{WSW}(\boldsymbol{x}) - \mathrm{WSW}(\boldsymbol{x}^*)| = O(\varepsilon)$.

*Proof.* A direct inference from Lemma B.1 and Lemma B.5. $\square$

### B.4.1 Proof of Lemma B.1

*Proof of Lemma B.1.* We observe that

$$\mathrm{OPT} - \mathrm{LNW}(\boldsymbol{x}) = \sum_{i \in [n]} B_i \left[ \log u_i(\boldsymbol{x}_i^*) - \log u_i(\boldsymbol{x}_i) \right]$$

Consider the Taylor expansion of $\log u_i(\boldsymbol{x}_i)$ and $u_i(\boldsymbol{x}_i)$:

$$\log u_i(\boldsymbol{x}_i) = \log u_i(\boldsymbol{x}_i^*) + \frac{1}{u_i(\boldsymbol{x}_i^*)}(u_i(\boldsymbol{x}_i) - u_i(\boldsymbol{x}_i^*))$$

$$- \frac{1}{2u_i(\boldsymbol{x}_i^*)^2}(u_i(\boldsymbol{x}_i) - u_i(\boldsymbol{x}_i^*))^2$$

$$+ O((u_i(\boldsymbol{x}_i) - u_i(\boldsymbol{x}_i^*))^3)$$

$$u_i(\boldsymbol{x}_i) = u_i(\boldsymbol{x}_i^*) + \frac{\partial u_i}{\partial \boldsymbol{x}_i}(\boldsymbol{x}_i^*)(\boldsymbol{x}_i - \boldsymbol{x}_i^*)$$

$$+ \frac{1}{2}(\boldsymbol{x}_i - \boldsymbol{x}_i^*)^T H(\boldsymbol{x}_i^*)(\boldsymbol{x}_i - \boldsymbol{x}_i^*) + O(\|\boldsymbol{x}_i - \boldsymbol{x}_i^*\|^3)$$

Notice that $\|\boldsymbol{x}_i - \boldsymbol{x}_i^*\| = \varepsilon$, we have

$$\log u_i(\boldsymbol{x}_i) = \log u_i(\boldsymbol{x}_i^*)$$

$$+ \frac{1}{u_i(\boldsymbol{x}_i^*)}[\frac{\partial u_i}{\partial x_i}(\boldsymbol{x}_i^*)(\boldsymbol{x}_i - \boldsymbol{x}_i^*) \cdots \varepsilon \text{ term} \tag{51}$$

$$+ \frac{1}{2}(\boldsymbol{x}_i - \boldsymbol{x}_i^*)^T H(\boldsymbol{x}_i^*)(\boldsymbol{x}_i - \boldsymbol{x}_i^*)] \cdots \varepsilon^2 \text{ term} \tag{52}$$

$$- \frac{1}{2u_i(\boldsymbol{x}_i^*)^2}\left(\frac{\partial u_i}{\partial \boldsymbol{x}_i}(\boldsymbol{x}_i^*)(\boldsymbol{x}_i - \boldsymbol{x}_i^*)\right)^2 \cdots \varepsilon^2 \text{ term} \tag{53}$$

$$+ O(\varepsilon^3)$$

We next deal with Equation (51) to Equation (53) separately.

**Derivation of Equation (51)**  Since $\boldsymbol{x}_i^*$ solves the buyer $i$'s problem, we must have

$$\frac{\partial u_i}{\partial x_i}(\boldsymbol{x}_i^*) = \lambda_i \boldsymbol{p}^* \tag{54}$$

where $\lambda_i$ is the Lagrangian Multipliers for buyer $i$.

We also know that $u_i(\boldsymbol{x}_i)$ is homogeneous with degree 1, by Euler formula, we derive

$$\langle \frac{\partial u_i}{\partial x_i}(\boldsymbol{x}_i), \boldsymbol{x}_i \rangle = u_i(\boldsymbol{x}_i) \tag{55}$$

Combine Equation (54) and Equation (55) and take $\boldsymbol{x}_i = \boldsymbol{x}_i^*$, we derive

$$\lambda_i \langle \boldsymbol{p}^*, \boldsymbol{x}_i^* \rangle = u_i(\boldsymbol{x}_i^*)$$

$$\lambda_i = \frac{u_i(\boldsymbol{x}_i^*)}{B_i}$$

$$\frac{\partial u_i}{\partial x_i}(\boldsymbol{x}_i^*) = \frac{u_i(\boldsymbol{x}_i^*)}{B_i}\boldsymbol{p}^*$$

Sum up over $i$ for Equation (51), we have

$$\sum_{i \in [n]} B_i \frac{1}{u_i(\boldsymbol{x}_i^*)} \frac{\partial u_i}{\partial x_i}(\boldsymbol{x}_i^*)(\boldsymbol{x}_i - \boldsymbol{x}_i^*)$$

$$= \boldsymbol{p} \sum_{i \in [n]} (\boldsymbol{x}_i - \boldsymbol{x}_i^*) \tag{56}$$

$$= 0 \cdots \text{by market clearance}$$

**Derivation of Equation (52) and Equation (53)**  Combining Equation (52) and Equation (53), we have

$$\frac{B_i}{2u_i(\boldsymbol{x}_i^*)}(\boldsymbol{x}_i - \boldsymbol{x}_i^*)^T H(\boldsymbol{x}_i^*)(\boldsymbol{x}_i - \boldsymbol{x}_i^*) - \frac{1}{2B_i}(\boldsymbol{x}_i - \boldsymbol{x}_i^*)^T(\boldsymbol{p}^*\boldsymbol{p}^{*T})(\boldsymbol{x}_i - \boldsymbol{x}_i^*)$$

$$= \frac{1}{2B_i}(\boldsymbol{x}_i - \boldsymbol{x}_i^*)^T(\frac{B_i^2}{u_i(\boldsymbol{x}_i^*)}H(\boldsymbol{x}_i^*) - \boldsymbol{p}^*\boldsymbol{p}^{*T})(\boldsymbol{x}_i - \boldsymbol{x}_i^*)$$

599    Denote $H_0(\boldsymbol{x}_i^*) = \frac{B_i^2}{u_i(\boldsymbol{x}_i^*)} H(\boldsymbol{x}_i^*) - \boldsymbol{p}^* \boldsymbol{p}^{*T}$, next we assert that $H_0(\boldsymbol{x}_i^*)$ is negative definite.

600    Since $H(\boldsymbol{x}_i^*)$ and $-\boldsymbol{p}^* \boldsymbol{p}^{*T}$ are negative semi-definite, $H_0(\boldsymbol{x}_i^*)$ must be negative semi-definite with
601    $\mathrm{rank}(H_0(\boldsymbol{x}_i^*)) \geq m - 1$.

602    Let $\lambda_1 \leq \lambda_2 \leq \cdots \leq \lambda_{m-1} < \lambda_m = 0$ be eigenvalues and $v_1, ..., v_n = \boldsymbol{x}_i^*$ be eigenvectors
603    of $H(\boldsymbol{x}_i^*)$. If $\mathrm{rank}(H_0(\boldsymbol{x}_i^*)) = m - 1$, it means that $\boldsymbol{x}_i^*$ has to be eigenvectors of $-\boldsymbol{p}^* \boldsymbol{p}^{*T}$ with
604    eigenvalue 0. However, we have $-\boldsymbol{p}^* \boldsymbol{p}^{*T} \boldsymbol{x}_i^* = -B_i \boldsymbol{p}^* \neq 0$, which leads to a contradiction.

605    Therefore, we have $\mathrm{rank}(H_0(\boldsymbol{x}_i^*)) = m$ and $H_0(\boldsymbol{x}_i^*)$ is negative definite, we denote $\lambda_1^i \leq ..., \leq$
606    $\lambda_n^i < 0$ as the eigenvalues of $H_0(\boldsymbol{x}_i^*)$, and $k$ as the universal lower bound for $|\lambda_n^i|$, then we have that,

$$\frac{1}{2}(\boldsymbol{x}_i - \boldsymbol{x}_i^*)^T H_0(\boldsymbol{x}_i^*)(\boldsymbol{x}_i - \boldsymbol{x}_i^*) \leq -\frac{k}{2}\varepsilon^2 \tag{57}$$

607    By combining Equation (56) and Equation (57), we have

$$\begin{aligned}
\mathrm{OPT} - \mathrm{LNW}(\boldsymbol{x}) &= -\sum_{i \in [n]} B_i \left[ \frac{1}{2B_i}(\boldsymbol{x}_i - \boldsymbol{x}_i^*)^T H_0(\boldsymbol{x}_i^*)(\boldsymbol{x}_i - \boldsymbol{x}_i^*) \right] + O(\varepsilon^3) \\
&\geq \frac{k}{2}\varepsilon^2 + O(\varepsilon^3) \\
&= \Omega(\varepsilon^2)
\end{aligned} \tag{58}$$

608    $\square$

### B.4.2   Proof of Lemma B.2

610    *Proof of Lemma B.2.* The proof is similar with Appendix B.4.1 by using Taylor expansion technique.
611    Before that, we first derive some identities.

612    By Roy's identity, we have

$$\frac{\partial \tilde{u}_i}{\partial p_j}(\boldsymbol{p}, B_i) = -x_{ij}^*(\boldsymbol{p}, B_i) \frac{\partial \tilde{u}_i}{\partial B_i}(\boldsymbol{p}, B_i)$$

613    Since $u(\boldsymbol{x}_i)$ is homogeneous with $\boldsymbol{x}_i$, it's easy to derive that

$$\frac{\partial \tilde{u}_i}{\partial B_i}(\boldsymbol{p}, B_i) = \frac{\tilde{u}_i(\boldsymbol{p}, B_i)}{B_i}$$

614    Above all,

$$\frac{\partial \tilde{u}_i}{\partial p_j}(\boldsymbol{p}, B_i) = -\frac{1}{B_i} x_{ij}^*(\boldsymbol{p}, B_i) \tilde{u}_i(\boldsymbol{p}, B_i)$$

615    Besides,

$$\begin{aligned}
\frac{\partial^2 \tilde{u}_i}{\partial p_j \partial p_k}(\boldsymbol{p}, B_i) =& \frac{1}{B_i^2} x_{ij}^*(\boldsymbol{p}, B_i) x_{ik}^*(\boldsymbol{p}, B_i) \tilde{u}_i(\boldsymbol{p}, B_i) \\
&- \frac{1}{B_i} \frac{x_{ij}^*(\boldsymbol{p}, B_i)}{\partial p_k} \tilde{u}_i(\boldsymbol{p}, B_i)
\end{aligned}$$

616    Next we consider the Taylor expansion,

$$\begin{aligned}
\log \tilde{u}_i(\boldsymbol{p}) =& \log \tilde{u}_i(\boldsymbol{p}^*) \\
&+ \frac{1}{\tilde{u}_i(\boldsymbol{p}^*)} [\frac{\partial \tilde{u}_i}{\partial \boldsymbol{p}}(\boldsymbol{p}^*)(\boldsymbol{p} - \boldsymbol{p}^*) \cdots \varepsilon \text{ term} \tag{59} \\
&+ \frac{1}{2}(\boldsymbol{p} - \boldsymbol{p}^*)^T H_p (\boldsymbol{p} - \boldsymbol{p}^*)] \cdots \varepsilon^2 \text{ term} \tag{60} \\
&- \frac{1}{2\tilde{u}_i(\boldsymbol{p}^*)^2} \left[ \frac{\partial \tilde{u}_i}{\partial \boldsymbol{p}}(\boldsymbol{p}^*)(\boldsymbol{p} - \boldsymbol{p}^*) \right]^2 \cdots \varepsilon^2 \text{ term} \tag{61} \\
&+ O(\varepsilon^3)
\end{aligned}$$

617    where $H_p$ is the Hessian matrix for $\tilde{u}_i(\boldsymbol{p})$.

618   **Derivation of Equation ([59](#))**  We have

$$\sum_{i\in[n]} B_i \frac{1}{\tilde{u}_i(\boldsymbol{p}^*)} \langle \frac{\partial \tilde{u}_i}{\partial \boldsymbol{p}}(\boldsymbol{p}^*), (\boldsymbol{p}-\boldsymbol{p}^*)\rangle$$

$$=\sum_{i\in[n]} \langle \boldsymbol{x}_i^*, (\boldsymbol{p}-\boldsymbol{p}^*)\rangle$$

$$=\langle \mathbf{1}, (\boldsymbol{p}-\boldsymbol{p}^*)\rangle \cdots \text{by market clearance}$$

$$=0 \cdots \text{by price constraints}$$

619   **Derivation of Equation ([60](#)) and Equation ([61](#))**  These expressions become

$$\frac{1}{2\tilde{u}_i(\boldsymbol{p}^*)}\big[\frac{1}{B_i^2}\tilde{u}_i(\boldsymbol{p}^*)\langle \boldsymbol{x}_i^*, \boldsymbol{p}-\boldsymbol{p}^*\rangle^2 - \frac{1}{B_i}\tilde{u}_i(\boldsymbol{p}^*)(\boldsymbol{p}-\boldsymbol{p}^*)^T(\frac{\partial x_{ij}^*}{\partial p_k}(\boldsymbol{p}^*, B_i))_{j,k\in[m]}(\boldsymbol{p}-\boldsymbol{p}^*)\big]$$

$$-\frac{1}{2\tilde{u}_i(\boldsymbol{p}^*)^2}\frac{\tilde{u}_i(\boldsymbol{p}^*)^2}{B_i^2}\langle \boldsymbol{x}_i^*, \boldsymbol{p}-\boldsymbol{p}^*\rangle^2$$

$$=\frac{1}{2B_i}(\boldsymbol{p}-\boldsymbol{p}^*)^T(\frac{\partial x_{ij}^*}{\partial p_k}(\boldsymbol{p}^*, B_i))_{j,k\in[m]}(\boldsymbol{p}-\boldsymbol{p}^*)$$

620   Summing up over $i$, we derive that

$$\mathrm{LFW}(\boldsymbol{p}) - \mathrm{OPT} = \sum_{i\in[n]} B_i \frac{1}{2B_i}(\boldsymbol{p}-\boldsymbol{p}^*)^T(\frac{\partial x_{ij}^*}{\partial p_k}(\boldsymbol{p}^*, B_i))_{j,k\in[m]}(\boldsymbol{p}-\boldsymbol{p}^*) + O(\varepsilon^3)$$

$$=\frac{1}{2}(\boldsymbol{p}-\boldsymbol{p}^*)^T H_X(\boldsymbol{p}-\boldsymbol{p}^*) + O(\varepsilon^3)$$

621  Since $\boldsymbol{p}^*$ gets the minimum of $\mathrm{LFW}(\boldsymbol{p})$, we must have that $H_X$ is positive semi-definite. Together
622  with $H_X$ has full rank, we know that $H_X$ is positive definite. Denote $\lambda_m$ as the minimum eigenvalues
623  of $H_X$, we have

$$\mathrm{LFW}(\boldsymbol{p}) - \mathrm{OPT} \geq \frac{\varepsilon^2 \lambda_m}{2} + O(\varepsilon^3)$$

$$=\Omega(\varepsilon^2)$$

624                                                            □

### B.4.3   Proof of Lemma [B.5](#)

626  *Proof of Lemma [B.5](#).*  Notice that

$$\mathrm{WSW}(\boldsymbol{x}) = \mathrm{WSW}(\boldsymbol{x}^*) + \sum_{i\in[n]} \langle \frac{\partial \mathrm{WSW}}{\partial \boldsymbol{x}_i}(\boldsymbol{x}_i^*), (\boldsymbol{x}_i - \boldsymbol{x}_i^*)\rangle + O(\varepsilon^2)$$

627  We have

$$\frac{\partial \mathrm{WSW}}{\partial \boldsymbol{x}_i}(\boldsymbol{x}_i^*)$$

$$=B_i \frac{\partial u_i}{\partial \boldsymbol{x}_i}(\boldsymbol{x}_i^*)$$

$$=B_i \frac{u_i(\boldsymbol{x}_i^*)}{B_i}\boldsymbol{p}^*$$

$$=c\boldsymbol{p}^*$$

628  Therefore,

$$\mathrm{WSW}(\boldsymbol{x}) = \mathrm{WSW}(\boldsymbol{x}^*) + \sum_{i\in[n]} c\langle \boldsymbol{p}^*, \boldsymbol{x}_i - \boldsymbol{x}_i^*\rangle + O(\varepsilon^2)$$

$$=\mathrm{WSW}(\boldsymbol{x}^*) + O(\varepsilon^2) \cdots \text{market clearance}$$

629  which completes the proof.

630                                                            □

## C  Additional Experiments Details

### C.1  More about baselines

**EG program solver (abbreviated as EG)**  We propose the first baseline algorithm EG. Recall the Eisenberg-Gale convex program(EG):

$$\max \quad \frac{1}{n} \sum_{i=1}^{n} B_i \log u_i(\boldsymbol{x}_i) \quad \text{s.t.} \quad \frac{1}{n} \sum_{i=1}^{n} x_{ij} = 1, \; x \geq 0. \tag{62}$$

We use the network module in pytorch to represent the parameters $\boldsymbol{x} \in \mathbb{R}_+^{n \times m}$, and softplus activation function to satisfy $x \geq 0$ automatedly. We use gradient ascent algorithm to optimize the parameters $\boldsymbol{x}$. For constraint $\frac{1}{n} \sum_{i \in [n]} x_{ij} = 1$, we introduce Lagrangian multipliers $\lambda_j$ and minimize the Lagrangian:

$$\mathcal{L}_\rho(\boldsymbol{x}; \boldsymbol{\lambda}) = -\frac{1}{n} \sum_{i \in [n]} B_i \log u_i(\boldsymbol{x}_i) + \sum_{j \in [m]} \lambda_j \left( \frac{1}{n} \sum_{i \in [n]} x_{ij} - 1 \right) \tag{63}$$

$$+ \frac{\rho}{2} \sum_{j \in [m]} \left( \frac{1}{n} \sum_{i \in [n]} x_{ij} - 1 \right)^2 \tag{64}$$

The updates of $\boldsymbol{\lambda}$ is $\lambda_j \leftarrow \lambda_j + \beta_t \rho \left( \frac{1}{n} \sum_{i \in [n]} x_{ij} - 1 \right)$, here $\beta_t$ is step size, which is identical with that in MarketFCNet. The algorithm returns the final $(\boldsymbol{x}, \boldsymbol{\lambda})$ as the approximated market equilibrium.

**EG program solver with momentum (abbreviated as EG-m)**  The program to solve is exactly same with that in EG. The only difference is that we use gradient ascent with momentum to optimize the parameters $\boldsymbol{x}$.

### C.2  More Experimental Details

Without special specification, we use the experiment settings as follows. All experiments are conducted in one RTX 4090 graphics cards using 16 CPUs or 1 GPU. We set dimension of representations of buyers and goods to be $d = 5$. Each elements in representation is i.i.d from $\mathcal{N}(0, 1)$ for normal distribution (default) contexts, $U[0, 1]$ for uniform distribution contexts and $Exp(1)$ for exponential distribution contexts. Budget is generated with $B(b) = ||b||_2$, and valuation in utility function is generated with $v(b, g) = \text{softplus}(\langle b, g \rangle)$, where $\text{softplus}(x) = \log(1 + \exp(x))$ is a smoothing function that maps each real number to be positive. $\alpha$ in CES utility are chosen to be 0.5 by default. MarketFCNet is designed as a fully connected network with depth 5 and width 256 per layer. $\rho$ is chosen to be 0.2 in Augmented Lagrange Multiplier Method and the step size $\beta_t$ is chosen to be $\frac{1}{\sqrt{t}}$. We choose $K = 100$ as inner iteration for each epoch, and training for 30 epochs in MarketFCNet. For *EG* and *EG-m* baselines, we choose the inner iteration $K = 1000$ when $n > 1000$ and $K = 100$ when $n \leq 1000$ for each epoch. Baselines are enssembled with early stopping as long as NG is lower than $10^{-3}$. Both baselines are optimized for 30 epochs in total.

We use Adam optimizer and learning rate $1e - 4$ to optimize the allocation network in MarketFCNet. When computing $\Delta \lambda_j$ in MarketFCNet, we directly compute $\Delta \lambda_j$ rather than generate an unbiased estimator, since it does not cost too much to consider all buyers for one time. For those baselines, we use gradient descent to optimize the parameters following existing works, and the step size is fine-tuned to be $1e + 2$ for $\alpha = 1, n > 1000$; $1e + 3$ for $\alpha < 1, n > 1000$ and 1 for $\alpha < 1, n \leq 1000$ and 0.1 for $\alpha = 1, n \leq 1000$ for better performances of the baselines. Since that Lagrangian multipliers $\lambda \leq 0$ will indicate an illegal Nash Gap measure, therefore, we hard code EG algorithm such that it will only return a result when it satisfies that the price $\lambda_j > 0$ for all good j. All baselines are run in GPU when $n > 1000$ and CPU when $n \leq 1000$.[1]

---

[1] We find in the experiments when market size is pretty large, baselines run slower on CPU than on GPU and this phenomenon reverses when market size is small. Therefore, the hardware on which baselines run depend on the market size and we always choose the faster one in experiments.

