# OpenReview forum: "Large-Scale Contextual Market Equilibrium Computation through Deep Learning"
_NeurIPS.cc/2024/Conference — Submitted to NeurIPS 2024_

### Official Review · Reviewer_5naV · 2024-07-09

**Soundness:** 2
**Presentation:** 2
**Contribution:** 3
**Rating:** 5
**Confidence:** 4

**Summary:**

This paper studies to how to use deep learning to solve large-scale contextual market equilibrium. This paper proposes MarketFCNet, a deep learning method for approximating market equilibrium. The paper propose an unbiased training loss and a metric called Nash Gap to quantify the gap between the learned allocation and the market equilibrium. Experiments on a synthetic game validates its effectiveness.

**Strengths:**

Originality: The paper propose a deep learning method to solve large-scale market equilibrium, which represents buyers and goods, and directly outputs the allocation. The application is novel.
Quality: The paper theoretically derives the loss function, and does some experimental analysis to validates the effectiveness of the propose method.
Clarity: The paper clearly defines the contextual market modeling problem.
Significance: Experiments validates that MarketFCNet are competitive with EG and achieve a much lower running time compared with traditional methods.

**Weaknesses:**

Quality: The paper does not prove the convergence of the training algorithm. The paper either does not show the training curve. The paper does not provide the implementation code of the algorithm.
Clarity: The paper is hard to follow. It is quite to hard to understand the meaning of each proposition.
Significance: The paper aims to solve the large scale contextual market equilibrium, and proposes a novel deep learning method  to approximate the equilibrium efficiently. However, the importance of the large scale contextual market equilibrium is not clear. I do not know how to apply the proposed method in real life.

**Questions:**

See the weakness.

**Limitations:**

Yes

---

> ### Author Rebuttal · Authors · 2024-08-07
>
> Thank you for your mindful comments! We will clarify some misunderstandings and address the concerns you have listed.
>
> * **Minor corrections**
>
> > The paper propose an unbiased training loss and a metric called Nash Gap to quantify the gap between the *learned allocation* and the market equilibrium.
>
> A market solution consists of allocations and prices of goods. Nash Gap is used to quantify the gap between the allocations and prices pair found by MarketFCNet and market equilibrium.
>
> > Experiments on a synthetic *game* validates its effectiveness.
>
> It's better to describe the problem as a "market" rather than a "game".
>
> * **About the concerns in Weaknesses**
>
> > The paper does not prove the convergence of the training algorithm.
>
> > The paper either does not show the training curve.
>
> We provide the loss curve of MarketFCNet and baselines in a PDF file attached to the *Author Rebuttal*. The loss curve shows that MarketFCNet algorithm converges well in all cases.
>
> > The paper does not provide the implementation code of the algorithm.
>
> We provide the implementation code in an anonymous link.
> As is required by NeurIPS2024, an external link is forbidden in the rebuttal period, thus the link to the codes is submitted to AC.
>
> > However, the importance of the large scale contextual market equilibrium is not clear.
>
> **The importance of market equilibrium**
>
> As is shown in the first paragraph of the paper, market equilibrium is one of the most important concept in microeconomic theory, see *Microeconomic theory* [1] for more details.
> Nobel Prize has also been awarded to the contributions of market equilibrium [2].
>
> [1] Andreu Mas-Colell, Michael Dennis Whinston, Jerry R Green, et al. Microeconomic theory, volume 1. Oxford University Press New York, 1995.
>
> [2] The Sveriges Riksbank Prize in Economic Sciences in Memory of Alfred Nobel 1972.
>
> **Why large scale contextual market equilibrium?**
>
> **Large scale**
>
> As is shown in the second paragraph of the paper, the real-world markets are happen to be large in the markets that are closedly related to everyone in the world, such as online shopping market and job market.
>
> **Contextual market**
> *In theory*, contextual representation is helpful to reduce the representation complexity of the model. In our contextual market example, traditional market specifies each good and each buyer with a value, which needs $O(nm)$ complexity; our method only need to specify the context of each buyer and each good, which needs only $O(n+m)$ complexity. Besides, contextual representation is helpful for training, which inspired our MarketFCNet method.
>
> *In practice*, contextual representation is more intrisic than representations in traditional market. Consider that one may ask following questions: buyer $i$ values $v_{ij}$ to the good $j$, how is $v_{ij}$ determined in reality? A natural answer is that $v_{ij}$ depends on the characteristics of buyer $i$ and good $j$, which is exactly represented by the contexts, $b_i$ and $g_j$.
> In fact, in real-world applications such as recommendation system, how much a person likes an item is predicted from the characteristics of the person and the item in the firm's view, which are exactly the concept of contexts in this paper. Although the recommendation system is not generally a market, we argue that the idea insight share a commonality.
>
> > I do not know how to apply the proposed method in real life.
>
> **A possible implementation of MarketFCNet in real life** We argue that MarketFCNet method can be implemented in real life. To do this, we first need to collect the contexts of all buyers and goods as well as valuation function of buyer context towards good context to construct a contextual market. As long as the model is constructed, the method can be directly implemented.
>
> The internet can helps us do this well since user records can be easily memorized and utilized. In many internet applications, such as recommendation system and advertisement auctions, firms collect the data of users and use representation learning to transform the data into contexts to better predict the valuations of them.
> We believe that the similar engineering can be used in the field of internet economy problem and digitalized real economy as well, e.g., market problem considered in this paper.
> Specifically speaking, online shopping platform can utilize the browing and purchase history of buyers to better predict the demand of some goods and provide the goods with guide prices. We also believe that in the future, everyone's preference (say, utility given different events) can be inferred by historical data, represented by context, leading to effecient usage for downstream economy problems.

---

### Official Review · Reviewer_Hy1m · 2024-07-12

**Soundness:** 3
**Presentation:** 4
**Contribution:** 3
**Rating:** 6
**Confidence:** 3

**Summary:**

This paper studies the computation of market equilibrium where there are a large number of buyers and the buyers and goods are represented by their contexts. It proposes a deep-learning method, termed MarketFCNet, to approximate the market equilibrium. The method outputs the good allocation by taking in the context embedding. It is trained on unbiased estimator of the objective function of EG-convex program using ALMM and is evaluated using a metric called Nash Gap. The method is validated by experimental results.

**Strengths:**

The paper is well-written and easy to understand. The motivation of the paper seems natural. The paper fills the gap of using deep learning for large scale market equilibrium computation, which can be promising for future study.

**Weaknesses:**

1. The proof of the unbiasedness of $\Delta \lambda_j$ and Lagrangian estimators in Sec 4.2 seems to be a bit hand-wavy. For example, should $b_i$’s be independent of each other? For a fixed $i$, is $b’_i$ an independent copy of $b_i$? It would be great if the authors could provide a formal (and more detailed) proof of the unbiasedness.

2. What is the effect of $k$ on the method performance? For example, if the dimension $k$ is very large, would the method fail to comprehend the context?

3. How to determine the architecture of allocation network? For example, can one use a Transformer or CNN as the allocation network?

Minor issues:

Line 164: It would be better to define $U(B)$ when introducing uniformly sampling to latter use.

Some equations are missing “.” or “,” at the end. Please fix those.

**Questions:**

See Weakness.

**Limitations:**

Yes.

---

> ### Author Rebuttal · Authors · 2024-08-07
>
> Thank you for your encouraging review! We appreciate that you affirmed our contributions. We will address the questions you listed.
>
> > 1. The proof of the unbiasedness ... is $b'_i$ an independent copy of $b_i$?
>
> We are sorry that our deductions make you confused. In our paper, we assume that there is a sampler that can generate a sequence of random variables independently and identically distributed (i.i.d.) from the distribution of $U(B)$. Therefore, all $\{b_i\}$s and $\{b'_i\}$s are i.i.d. samples of the distribution $U(B)$.
>
> > 1. (continued) It would be great if the authors could provide a formal (and more detailed) proof of the unbiasedness.
>
> We provide a formal proof in a PDF file.
> As is required by NeurIPS2024, only figure and table can appear in the submitted PDF file, and external link is forbidden in the rebuttal period, thus the link to the PDF file is submitted to AC.
>
>
> > 2. What is the effect of $k$ on the method performance? For example, if the dimension $k$ is very large, would the method fail to comprehend the context?
>
> We presume that you refer $k$ as the context dimension of buyers and goods. If $k$ is very large, we believe that, our method would fail (in the sense of having no performance advantage over traditional methods) in the worst case, and still work in the "good case", here we meant "good case" as the cases that are likely to appear in the real world. It is mainly because a widely accepted assumption that many high-dimensional real-world problem has an intrinsic low-dimensional structure (this can be evidenced by the success of autoencoder and its variants [1]). If the context is high-dimensional, then we can use extra method (e.g. autoencoder) to reduce the context to be low-dimensional, followed by MarketFCNet with transformed low-dimensional contexts.
> Therefore, high-dimensional contexts do not hurt the comprehensiveness of model too much.
>
> However, we argue that $k$ is usually small in the real world scenario, which certifies that the model will work on the real world case. There are many experimental works that corporates context into network design. As an example, the context dimension in Duan et al. [2], is not large in general.
> On the other hands, in many settings when utilities of buyers take special forms, such as linear utility, Leontief utilities, Cobb-Douglas utilities and CES utilities, etc. (See *Algorithmic Game Theory* [3] $\S$ 6.1.5 for more details), describing a utility of buyer need only $m$ parameters. Therefore, taking $k=m$ is always enough in these cases.
>
> [1] Kramer, Mark A. "Nonlinear principal component analysis using autoassociative neural networks." AIChE journal 37.2 (1991): 233-243.
>
> [2] Zhijian Duan, Jingwu Tang, Yutong Yin, Zhe Feng, Xiang Yan,Manzil Zaheer, and Xiaotie Deng. A context-integrated transformer-based neural network for auction design. In International Conference on Machine Learning, pages 5609–5626. PMLR, 2022.
>
> [3] Noam Nisan, Tim Roughgarden, Eva Tardos, and Vijay V Vazirani. Algorithmic game theory, 2007.
>
> > 3. How to determine the architecture of allocation network? For example, can one use a Transformer or CNN as the allocation network?
>
> In this paper, MarketFCNet adopts MLP architecture, and we find that the performance of MLP is good enough.
>
> Actually, MarketFCNet can work with arbitrary network architecture, as long as the network takes buyers and goods contexts as input and output corresponding allocations, see the *MarketFCNet* module in Figure 1 in original paper. This showcases the flexibility of MarketFCNet.
>
> In this sense, it's feasible to introduce Transformer or CNN architecture to MarketFCNet. For example, if the buyers/goods/contexts have locality or spatial structure, then CNN structure might be helpful for better performance; if the buyers/goods/contexts are sequential information and with variable length, then a Transformer architecture might be helpful for this case.
>
> We believe that a specifically designed architecture might result in better performance and lower running cost compared with MLP architecture in some situations, which we leave as a promising future work.
>
> > Line 164: It would be better to define $U(B)$ when introducing uniformly sampling to latter use.
>
> > Some equations are missing “.” or “,” at the end. Please fix those.
>
> Thank you for your careful elaboration! We will fix these issues in the final version.

---

> > ### Comment · Reviewer_Hy1m · 2024-08-08
> >
> > Thank you for your clarifications! Since I do not have access to the PDF file for the proof of unbiasedness, I cannot check its correctness directly. Other justifications are satisfactory. I will keep my rating (Weak Accept).

---

### Official Review · Reviewer_m3RG · 2024-07-13

**Soundness:** 3
**Presentation:** 3
**Contribution:** 3
**Rating:** 5
**Confidence:** 1

**Summary:**

The submission is not in your area and extends beyond my current expertise (from theory and applications to specific tasks and methods).

**Strengths:**

The submission is not in your area and extends beyond my current expertise (from theory and applications to specific tasks and methods).

**Weaknesses:**

The submission is not in your area and extends beyond my current expertise (from theory and applications to specific tasks and methods).

**Questions:**

The submission is not in your area and extends beyond my current expertise (from theory and applications to specific tasks and methods).

**Limitations:**

The submission is not in your area and extends beyond my current expertise (from theory and applications to specific tasks and methods).

---

### Official Review · Reviewer_84Ug · 2024-07-18

**Soundness:** 3
**Presentation:** 3
**Contribution:** 3
**Rating:** 5
**Confidence:** 3

**Summary:**

This paper proposes a deep learning-based method called MarketFCNet to efficiently compute market equilibrium in large-scale contextual markets, where buyers and goods are represented by their contexts. The key idea is to parameterize the allocation of each good to each buyer using a neural network, and optimize the network parameters through an unbiased estimation of the objective function. This approach significantly reduces the computation complexity compared to traditional optimization methods, making it suitable for markets with millions of buyers. Experimental results demonstrate that MarketFCNet delivers competitive performance and much faster running times as the market scale expands, highlighting the potential of deep learning for approximating large-scale contextual market equilibrium.

**Strengths:**

The deep learning-based approach, MarketFCNet, can efficiently approximate the market equilibrium in large-scale contextual markets by parameterizing the allocation using a neural network. This significantly reduces the computation complexity compared to traditional methods.

The ability to handle large-scale markets with millions of buyers makes this approach highly relevant for real-world scenarios, such as job markets, online shopping platforms, and ad auctions with budget constraints.

The paper introduces a new metric called Nash Gap to quantify the deviation of the computed allocation and price pair from the true market equilibrium, providing a meaningful way to evaluate the approximated solutions.

**Weaknesses:**

The deep learning-based approach is inherently less interpretable compared to traditional optimization methods. Exploring ways to improve the interpretability of the learned allocation function, such as incorporating domain-specific constraints or incorporating interpretable components, could enhance the practical usability of the method.

The paper does not discuss potential overfitting issues that may arise when training the MarketFCNet model, especially in settings with a large number of parameters. Incorporating appropriate regularization techniques and cross-validation strategies could help mitigate overfitting and improve the generalization performance.

The paper assumes that the contexts of buyers and goods are homogeneous and can be directly used as inputs to the neural network. Extending the approach to handle heterogeneous context representations, potentially by incorporating feature engineering or meta-learning techniques, could increase the applicability to more diverse market scenarios.

**Questions:**

From a technical perspective, what are the main strengths of the proposed deep learning-based approach, MarketFCNet, for computing market equilibrium in large-scale contextual markets?

What novel evaluation metric is introduced in this paper to assess the quality of the approximated market equilibrium solutions, and how does it contribute to the methodological advancements?

Given the deep learning-based nature of the approach, how might the authors address potential issues like overfitting or the ability to handle heterogeneous context representations in a more robust manner?

---

> ### Author Rebuttal · Authors · 2024-08-07
>
> Thank you for your detailed questions! We will address the questions and concerns you have listed.
>
> * **About the concerns in Weaknesses**
>
> > Exploring ways to improve the interpretability of the learned allocation function, such as incorporating domain-specific constraints or incorporating interpretable components, could enhance the practical usability of the method
>
> We have two constraints in our problem, which is shown in the expression between line 164 and 165. The second constraint $\boldsymbol{x}_ \theta (\boldsymbol{b},\boldsymbol{g}) \ge 0$ means that the good can only be non-negatively allocated. This constraint is incorporated in MarketFCNet by an element-wise softplus operation on the output, where $softplus(x) = ln(1 + e^x)$ is a differential function transforms real numbers to positive.
>
> For the first constraint $\mathbb{E}_b[x_ \theta (b,g_j)] \equiv 1$, we do not incorporate this constraint in MarketFCNet, since an exact computation of the expectation costs too much in MarketFCNet. However, we agree that exploring other ways to incorporate this constraint is a good direction for future work.
>
> > The paper does not discuss potential overfitting issues ... could help mitigate overfitting and improve the generalization performance.
>
>
> We do not find the overfitting phenomenon in our experiments.
> Our task is to compute (approximate) the market equilibrium of a large market. The performance of this task is measured by Nash Gap, which is a theoretically guaranteed measure of our task. In our experiments, MarketFCNet perform well as we test the performance through the Nash Gap measure.
>
>
> > Extending the approach to handle heterogeneous context representations, potentially by incorporating feature engineering or meta-learning techniques, could increase the applicability to more diverse market scenarios.
>
> We agree with your point. In fact, there is always an easy way to transform the heterogeneous context representations to a homogeneous one. For example, if buyer $i$ has context $b_i \in \mathbb{R}^{k_ i}$ for $i\in\{1,2\}$ and $k_1 < k_2$. We could let $b'_i = (b_i, 0^{k_2 - k_1})$ such that $b'_1$ and $b_2$ are homogeneous, where $0^{k_2 - k_1}$ represents a zero vector with dimension $k_2 - k_1$. Besides, this transformation has no representation loss about the context.
>
> * **Response to the questions**
>
> > From a technical perspective, what are the main strengths of the proposed deep learning-based approach, MarketFCNet, for computing market equilibrium in large-scale contextual markets?
>
> The main strength of MarketFCNet is that the optimization cost is much lower than traditional methods.
> As is analyzed in the second and fourth paragraph in the Introduction part, traditional methods take at least $O(nm)$ cost to do one optimization step (For more information, please refer to Gao and Kroer [1]). However, MarketFCNet only take $O(Km)$ cost, where $K$ is the computational cost through one network call. The network does not depends on $n$ and $m$, thus $K$ can be seen as a constant. Since the arbitrarily large and potentially infinite $n$, MarketFCNet is theoretically more computational efficient compared with traditional methods.
>
> [1] Yuan Gao and Christian Kroer. First-order methods for large-scale market equilibrium computation. Advances in Neural Information Processing Systems, 33:21738–21750, 2020.
>
> > What novel evaluation metric is introduced in this paper to assess the quality of the approximated market equilibrium solutions, and how does it contribute to the methodological advancements?
>
> We introduce Nash Gap in this paper, which is a novel evaluation metric of the approximated market equilibria. The introduce of Nash Gap does not directly help for methodological advancements. However, the analysis of Nash Gap help us understand the linear structure of the market equilibrium, which guide us to do linear transformation to arbitrary approximated market solutions when the assumption of Nash Gap does not hold generally.
>
> > Given the deep learning-based nature of the approach, how might the authors address potential issues like overfitting or the ability to handle heterogeneous context representations in a more robust manner?
>
> Please see our explainations about the overfitting issues and heterogeneous context representations above.

---

### Author Rebuttal · Authors · 2024-08-07

To reviewer 5naV: the training curve is provided in the attached PDF file.

---

### Decision · Program_Chairs · 2024-09-25

**Decision:**

Reject

**Comment:**

Summary:
The paper addresses a market cap equilibrium problem, where the authors assume a (very) large population of buyers. After revisiting the current literature to form equilibrium as a convex optimization problem, the authors propose MarketFCNet, which is more attractive when the number of buyers is high. To evaluate their approach, the authors introduce a metric called "Nash Gap." Finally, they conduct a series of experiments to demonstrate the effectiveness of their approach.

Review:
The paper contributes to the differentiable economics literature.